# Outdoor Residual Insecticide Spraying (ODRS), a New Approach for the Control of the Exophilic Vectors of Human Visceral Leishmaniasis: *Phlebotomus orientalis* in East Africa

**Dia-Eldin A. Elnaiem**[1]*, **Osman Dakein**[2,3], **Ahmed Mohammed-Ali Alawad**[4,5], **Bashir Alsharif**[6], **Altayeb Khogali**[5], **Tayseer Jibreel**[5], **Omran F. Osman**[2], **Hassan Has'san**[4,5], **Atia Mohamed Atia**[7], **Mousab Elhag**[8], **Margriet Den Boer**[9], **Koert Ritmeijer**[9], **Caryn Bern**[10], **Jorge Alvar**[11], **Noteila Khalid**[12], **Orin Courtenay**[13]*

**1** Department of Natural Sciences, University of Maryland Eastern Shore, MD, United States of America, **2** Department of Zoology, Faculty of Science, University of Khartoum, Sudan, **3** Kala azar Research Centre, Faculty of Medicine and Health Sciences, University of Gedarif, Gedarif, Sudan, **4** Ministry of Health, Gedarif state, Sudan, **5** Blue Nile Health Institute, Gezira University, Wad Medani, Sudan, **6** Departamento de Entomologia, CPqAM, Fundação Oswaldo Cruz, Recife, Brasil and Dept of Medical Entomology, National Public Health Laboratory, Ministry of Health, Sudan, **7** ASCEND Program, Khartoum, Sudan, **8** Director, Directorate of Communicable Diseases, Federal Ministry of Health, Khartoum, Sudan, **9** Medecins Sans Frontieres, Amsterdam, the Netherlands, **10** University of California San Francisco, San Francisco, California, United States of America, **11** Drugs for Neglected Diseases *initiative*, Geneva, Switzerland, **12** Department of Zoology, Ibn Sina University, Khartoum, Sudan, **13** Zeeman Institute and School of Life Sciences, University of Warwick, Coventry, CV4 7AL, United Kingdom

* daelnaiem@umes.edu (DEAE); orin.courtenay@warwick.ac.uk (OC)

## Abstract

Visceral Leishmaniasis (VL) due to *Leishmania donovani* is a neglected protozoan parasitic disease in humans, which is usually fatal if untreated. *Phlebotomus orientalis*, the predominant VL vector in East Africa, is a highly exophilic/exophagic species that poses a major challenge to current Integrated Vector Management (IVM). Here we report results of pilot studies conducted in rural villages in Gedarif state, Sudan, to evaluate outdoor residual spraying of 20mg active ingredient (a.i.) /m$^2$ deltamethrin insecticide applied to the characteristic household compound boundary reed fence and to the outside of household buildings (Outdoor Residual Insecticide Spraying, ODRS), and as an alternative, spraying restricted to the boundary fence only (Restricted Outdoor Residual Insecticide Spraying, RODRS). Four to six clusters of 20 households were assigned to insecticide treatments or control in three experiments. Changes in sand fly numbers were monitored over 2,033 trap-nights over 43–76 days follow-up in four sentinel houses per cluster relative to unsprayed control clusters. Sand fly numbers were monitored by sticky traps placed on the ground on the inside ("outdoor") and the outside ("peridomestic") of the boundary fence, and by CDC light traps suspended outdoors in the household compound. The effects of ODRS on sand fly numbers inside sleeping huts were monitored by insecticide knockdown. After a single application, ODRS reduced *P. orientalis* abundance by 83%-99% in outdoor and peridomestic

**Data Availability Statement:** All relevant data are within the manuscript and its Supporting Information files.

**Funding:** The authors gratefully acknowledge financial support from Department of International Development (DFID), the UK AID programme, on Tackling Visceral Leishmaniasis in East Africa, through the KalaCORE consortium (contract PO 6361) (www.kalacore.org). DFID covered all field expenses, data and data collection and analysis. OC acknowledges support from the Global Challenges Research Fund, Networks in Vector Borne Disease Research, BBSRC grant BB/R005362/1. The funders had no role in study design, data collection and analysis, decision to publish, or preparation of the manuscript. The views expressed do not necessarily reflect the UK government's official policies.

**Competing interests:** The authors have declared that no competing interests exist.

trap locations. ODRS also reduced numbers of *P. orientalis* found resting inside sleeping huts. RODRS reduced outdoor and peridomestic *P. orientalis* by 60%-88%. By direct comparison, RODRS was 58%-100% as effective as ODRS depending on the trapping method. These impacts were immediate on intervention and persisted during follow-up, representing a large fraction of the *P. orientalis* activity season. Relative costs of ODRS and RODRS delivery were $5.76 and $3.48 per household, respectively. The study demonstrates the feasibility and high entomological efficacy of ODRS and RODRS, and the expected low costs relative to current IVM practises. These methods represent novel sand fly vector control tools against predominantly exophilic/exophagic sand fly vectors, aimed to lower VL burdens in Sudan, with potential application in other endemic regions in East Africa.

## Author summary

*Phlebotomus orientalis* is the predominant vector of visceral leishmaniasis (VL, kala azar) in Sudan and other countries of East Africa, where the disease causes high morbidity and mortality. This sylvatic sand fly species is abundant in wild habitats characterized by presence of black cotton soil and vegetation dominated by *Balanites aegyptiaca* and/or *Acacia seyal* trees. In villages, the vector bites people in the household yard and in nearby peridomestic locations, exhibiting limited indoor resting behaviour. The marked exophagic and exophilic behaviours of *P. orientalis* represent a profound challenge for VL control by excluding indoor residual spraying of insecticides (IRS) and compromising the efficacy of insecticide-impregnated bednets (ITNs). In this study, we evaluated the entomological efficacy of residual pyrethroid applied outdoors to household boundary fences and the exterior walls of household huts (outdoor residual insecticide spraying, ODRS), to reduce the abundance of *P. orientalis* inside and outside houses. We also evaluated the entomological impact of a restricted outdoor residual insecticide spraying (RODRS), whereby insecticide was applied only to the boundary fence. The study was carried out in June 2016-June 2017 in Jebel-Algana and Umsalala villages, Gedarif state, eastern Sudan, which are highly endemic for VL. The results showed that a single ODRS application of 20mg a.i. /m$^2$ 2.8% deltamethrin provided average reductions of 83%-99% in outdoor and peridomestic *P. orientalis* sand fly numbers relative to unsprayed control clusters. RODRS reduced outdoor and peridomestic *P. orientalis* by 60%-88%. The average cost of ODRS and RODRS per household were $5.76 and $3.48, respectively. The costs of these community-based control measures were substantially lower than the costs of LLINs, which is the only evidence-based tool used to protect against VL in the area. Future studies should evaluate the impact of ODRS/RODRS transmission of VL incidence in endemic villages and in seasonal agricultural farms.

## Introduction

Standard methods to combat hematophagous arthropod vectors of infectious diseases include–among other possibilities- indoor residual spraying of insecticides (IRS) and use of insecticide-treated bednets (ITNs), the success of which largely rely on insecticide-susceptible vector populations showing endophilic and/or endophagic blood-feeding behaviours. Sustainable methods to control exophilic/exophagic vectors, by contrast, are lacking, but key in the fight against many of the so-called "tool-deficient" Neglected Tropical Diseases (NTDs)

[1,2,3]. One such challenge includes the Phlebotomine sand fly vectors of the protozoan parasites *Leishmania* causing human visceral leishmaniasis (VL, also known as kala azar) which occurs in East Africa, the Indian sub-continent, the Americas, the Mediterranean region, and Central and Eastern Asia. Globally, an underreported estimated 31,000–90,000 new cases are reported each year [4,5].

Visceral leishmaniasis, caused by *L. donovani* is a serious public health problem in E. Africa with some of the highest case incidences worldwide, resulting in severe epidemics in the region [4,5,6,7,8,9,10,11,12], and exerting tremendous social and economic burdens on afflicted populations [13,14]. The symptoms of systemic clinical VL development include fever, weight loss, fatigue, splenomegaly and hepatomegaly. VL is usually fatal within two years if not treated, and there is no human vaccine.

In East Africa, the prominent VL vector is *Phlebotomus orientalis*, which thrives in remote woodlands, villages, and peridomestic habitats, characterized by the presence of black cotton soil and *Acacia seyal* and *Balanites aegyptiaca* trees [13,15,16,17,18,19,20,21]. The vector is highly exophilic and exophagic, rarely captured inside household buildings [22,23,24,25,26,27,28,29] with villagers being exposed outdoors in the household compound and in the peridomestic surroundings. *Phlebotomus orientalis* is a highly seasonal species. In Sudan and neighbouring countries of East Africa, the vector starts to appear in small numbers following the rains in October, reaching peak abundance in March-June, before disappearing at the onset of the heavy rains at the beginning of July [15,16,17,20,21,25]. Studies on the seasonality of infection rates of *L. donovani* in *P. orientalis*, and longitudinal studies on the incidence of VL in Sudan, demonstrate that the transmission of the parasite occurs between March and June [7,16,30,31].

The highest burden of clinical VL in East Africa occurs in Sudan and South Sudan [4,5] with highest prevalences (43%-70%) reported in children <15-year-old [4,12,32]. In Sudan, 90% of cases occur in Gedarif state, eastern Sudan, where 51,200 patients were registered receiving treatment against VL between 2002–2015 [12], and the case incidence ranges between 26–37/1000/year (2014–2017 MoH data) resulting in 2,000–7,000 newly reported cases per year [12,33].

Integrated Vector Management (IVM) recommendations in the region include IRS using pyrethroid or Bendiocarb–(a carbamate)- twice per year, and provision of long-lasting insecticide-impregnated bednets (LLINs) to villagers. These approaches have been shown to impact on the local mosquito vectors of malaria e.g. *Anopheles arabiensis* [34], which bites and rest inside huts during the rainy season. In contrast, IRS has little potential in control of exophilic and exophagic species such as *P. orientalis*, which is unlikely to land on unsprayed or sprayed indoor wall surfaces [35]. Furthermore, the effectiveness of LLINs against *P. orientalis* is compromised by the habit of sleeping outdoors in the uncomfortably hot weather during the sand fly season (requiring setting up the bednets each night), and where a large fraction of *P. orientalis* bites occur early evening, before bedtime [9]. Indeed, the propensity to sleep outdoors may be a contributory factor to the marked exophagic behaviour of *P. orientalis* in Sudan and elsewhere.

To provide privacy to this outdoor sleeping area, each household yard is surrounded by 1.5–2 m tall fence made of tightly thatched reed, thus separating the outdoor yard from the immediate peridomestic area. In a recent investigation of microhabitat distribution of *P. orientalis* in 6 villages in Gedarif state, we captured lower numbers of the vector inside houses than their immediate surrounding; a result indicating that these boundary fences act as physical barriers, reducing numbers of *P. orientalis* entering the household yard and sleeping huts from the peridomestic habitats [36].

In pursuit of feasible alternative community-wide vector control methods, we investigated the impact on vector abundance of outdoor coverage with residual pyrethroid insecticide.

Pilot studies were designed specifically to test the entomological efficacy of (i) outdoor residual spraying of the walls of household huts and the boundary fence (ODRS), to reduce indoor, outdoor and peridomestic *P. orientalis* abundance; (ii) to compare the efficacy of ODRS to an alternative insecticide application restricted to spraying boundary fences alone (RODRS); and (iii) to estimate the relative costs of ODRS and RODRS.

## Methods

### Ethics statement

Insecticide spraying was approved by the district (Mahalia) health authorities of Gureisha and Rahad. Following a full explanation of the spraying procedures and the study objectives, informed written consent was obtained from the heads of all recruited households, and the head of the popular committee of the village. The study did not involve sampling human subjects.

### Study sites

The study was carried out in the rural villages of Jebel-Algana (13.6294444, 36.13666667) and Umsalala (12.85055556, 35.1822222) in Gedarif State, eastern Sudan, located within 70km of the northern Sudanese-Ethiopian border (Fig 1). Jebel-Algana is in Eastern Gureisha Mahlia (district), near the Atbara River, 129 km distance from Umsalala village [Western Galabat Mahalyia], located on the eastern bank of the River Rahad.

The villages are representative of this hyperendemic region of VL in eastern Sudan [37], experiencing high transmission rates. The stability in *P. orientalis* biting rates in the region is indicated by the consistent human VL incidence rates in previous and current years [31,38,39,40]

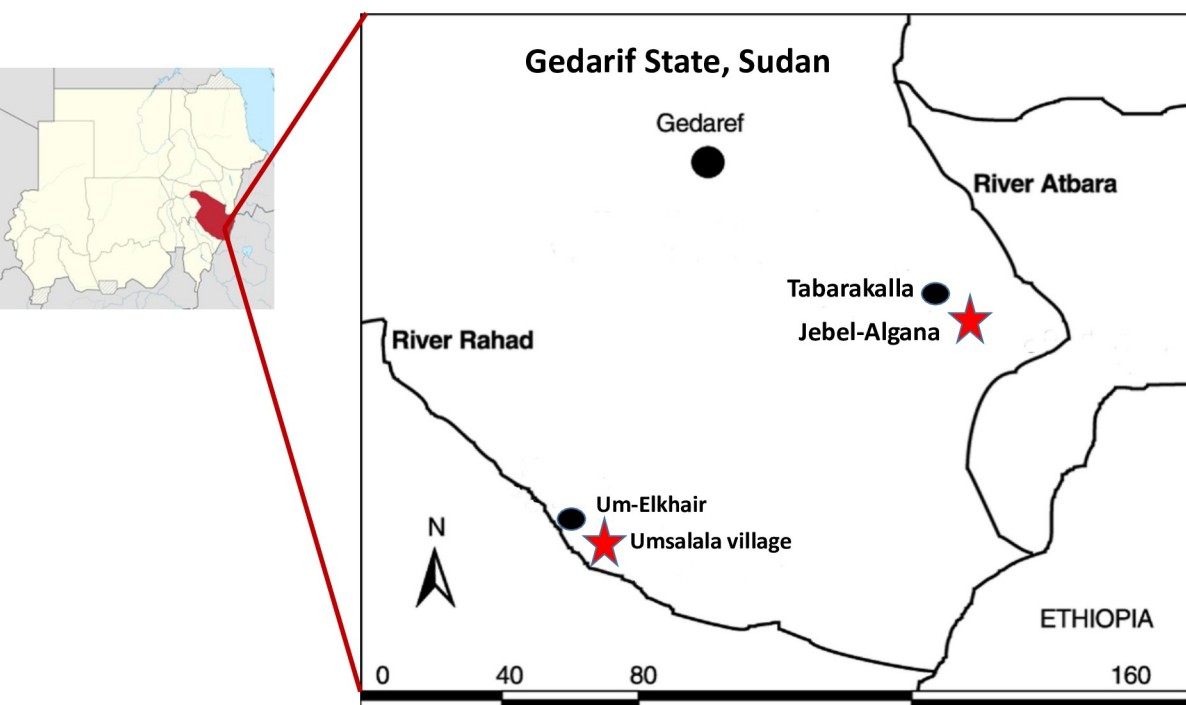

**Fig 1. Map of Gedarif state, Sudan, showing location of Jebel-Algana and Umsalala study villages in relation to Tabarakallah and Umelkher kala azar treatment centers.**

Both villages lie in an area that was previously covered by dense woodlands of *Acacia seyal* and *Balanites aegyptiaca*, but now greatly reduced by mechanized seasonal plantations of sorghum, sesame, pearl millet ("dokhun") and groundnuts, leaving remnant thickets of *B. aegyptiaca*, *A. seyal* and *Zyzyphus spina christi* trees in and around the villages. The soil is predominantly black cotton soil (chromic vertisols), interrupted by sandy and silt soils ("azaza"). During the dry season, the black cotton soil shrinks; creating deep cracks which are thought to harbour several species of sand fly including *P. orientalis* [15,18,41]. The climate is characterised as tropical continental, with annual rainfall of approximately 600–800 mm. The year is divided into a hot dry season (March-May), a moderately warm rainy season (June-October), and a warm dry winter (November- February). The regional mean temperature and relative humidity respectively are 32.3˚C and 25.9% in the hot dry season, 26.2˚C and 54.4% in the rainy season, and 27.0˚C and 21.9% in the winter. *Phlebotomus orientalis* is active for 3–4 months between March and June, being most abundant at the end of the hot dry season, after which the abundance rapidly declines at the onset of the heavy seasonal rains in June/July [15,17].

The resident populations of Jebel-Algana (3,955 inhabitants) and Umsalala (2,710 inhabitants) are predominantly agrarian comprising a number of ethnic groups following waves of migration for agricultural work between the early 1950s to the 1980s from the Darfur region, western Sudan [34]. Household compounds are typically surrounded by a 1.5–2 metre tall boundary fence made of tightly woven thatched reed *Cymbogon nyrvatus*, primarily to provide privacy (Fig 2A). Compounds contain an average of three (range: 1–4) huts ("rakobas"), one used for keeping belongings and sleeping in cooler months; the others used for cooking, storage, or daytime shelter. These huts are predominantly constructed of a wooden frame thatched with grass on the walls and roof. Most households maintain a variable small number of mixed livestock (chickens, sheep, goats, cattle, and donkeys); villagers rarely keep dogs or cats. Due to the excessive heat, people sleep outside in the compound throughout October–May, and on hotter dry nights of June-September [9,38]. They sleep inside huts only during the rains (June-

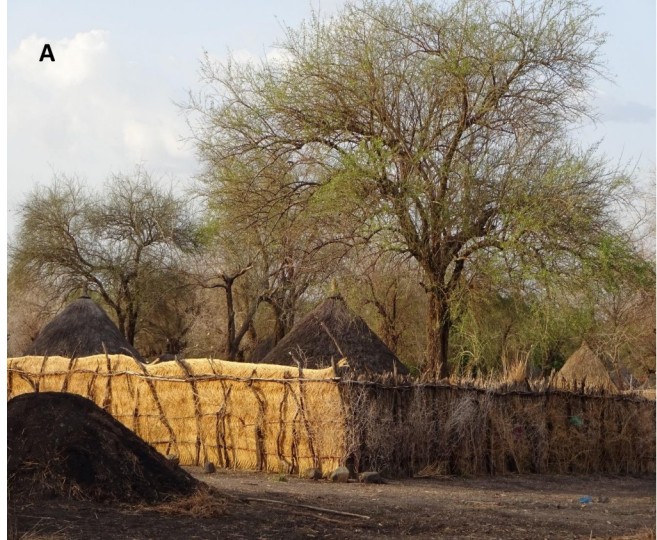 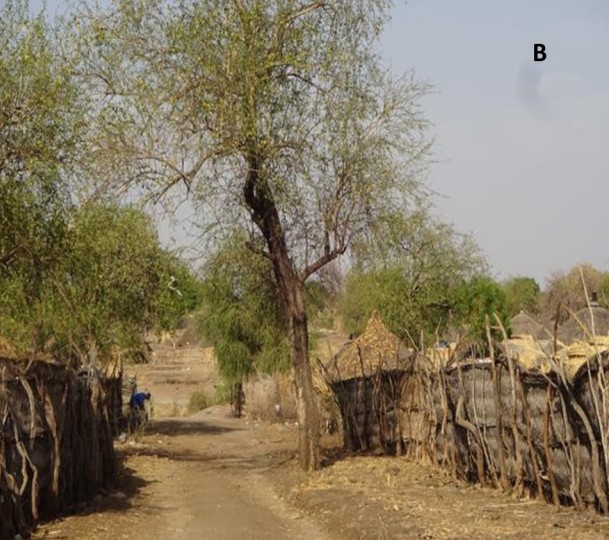

**Fig 2. Images of typical household layout in VL endemic villages in Gedarif state, eastern Sudan (this example in Jebel-Algana study village).** Each house consists of 2–4 thatched grass huts surrounded by a tall thatched reed fence that provides privacy. (A) *Acacia* and *Balanites* trees, associated with *P. orientalis* vectors, are commonly present within the property boundary, and (B) in the narrow earthen streets that separate household blocks. During the VL transmission season people sleep outside in within the household yard.

October) and in the winter (November- February). The villages are designed on a grid where blocks of contiguous houses are separated from neighbouring blocks by well-demarcated earthen tracks (Fig 2B).

### Village, cluster and sentinel house recruitment

**Villages.**   The two study villages were selected as a convenience sample from many possible VL endemic villages along the Atbara and Rahad rivers, on the criteria that within 1–2 years prior to the study there was evidence of active *L. donovani* transmission; an abundance of *P. orientalis* in the villages; and they were geographically located within 5 km from the field operations centres in Tabarakallah and Umelkher, respectively, to aid in logistical requirements. Jebel-Algana and Umsalala villages had 17 and 55 human VL cases recorded by the nearby Kala-azar treatment centres in Tabarakallah (data for 2015–2016) and Umelkher (data for 2016–2017), respectively. *Phlebotomus orientalis* presence was demonstrated by sand fly capture conducted in 2015–2016 [36]. During the current study, as in past entomological surveys, *P. orientalis* densities were generally higher in Umsalala than in Jebel-Algana [36].

**Clusters.**   Clusters were defined as 20 contiguous households. Ten clusters were identified in Jebel-Algana and Umsalala villages for possible recruitment into the study. The selection criteria included that clusters within villages were separated from neighbouring clusters by a distance of >300 metres to avoid possible contamination by sand fly dispersal [42,43,44]; that clusters showed relatively typical vegetation, topography, soil type and that each of the household compounds was surrounded by the characteristic reed fence.

Following consultation with village leaders, and informed written consent from all householders, four clusters in Jebel-Algana village (Experiment 1), and four clusters in Umsalala villages (Experiment 2) were recruited. The same four clusters used in Experiment 1 in 2016, plus two additional clusters in the same village, Jebel-Algana, were recruited for Experiment 3 in 2017. Due to the grid design of houses within villages, each household faced an earthen street on one side, and bordered neighbouring household compounds on the other sides. Therefore, to maximise proximity of households within clusters, each cluster comprised one linear block of 10 contiguous houses separated from the second block of 10 contiguous houses by a narrow earthen street (2–3 metres wide).

**Sentinel houses.**   Four sentinel households per cluster were identified for monitoring intervention effects on sand fly abundance. The four households were located in the centre of the cluster i.e. two households in the centre of each block of ten houses. The portion of the boundary fence facing the earthen street were the focus for monitoring sand fly numbers as described below.

### Study design

The study comprised of three experiments: Experiment 1, conducted between 9th and 16th June 2016, was the initial scoping study to assess the potential and feasibility of ODRS to reduce *P. orientalis* abundance inside sleeping huts, outdoors in the household compound, and outside the compound boundary fence. Experiment 2 was designed to evaluate the residual effectiveness of a single application of ODRS against outdoor and peridomestic sand fly numbers compared to controls followed-up over 43 days post intervention. Experiment 3 similarly aimed to assess the residual effectiveness against *P. orientalis* numbers of a single ODRS application, compared to single restricted ODRS (RODRS) application; over 76 follow-up days post intervention. Experiments 2 and 3 were conducted between 2nd May and 16th June 2017, and between 31st March and 20th June 2017, respectively. Following the baseline pre-intervention sample, each

intervention was applied on a single occasion per experiment, on 13[h] June 2016 (Experiment 1), 4[th] May 2017 (Experiment 2), and 5[th] April 2017 (Experiment 3).

## Intervention randomization

The enrolled clusters were randomised to one of the two interventions (Experiments 1 & 2), or to one of three interventions (Experiment 3), by an independent blinded local observer who selected folded slips of paper with cluster names written on them from a container. The interventions were assigned alternatively in order that cluster names were drawn to achieve two clusters per treatment in each Experiment 1–3. All houses within a cluster received the same intervention. No significant differences in pre-intervention fly numbers captured by Sticky traps (ST) or CDC light trap (LT) were detected when controlling for study design variables (see below), suggesting that the randomisation process for the three Experiments was successful ($z < 2.01$, $p > 0.347$).

## The interventions

ODRS comprised of spraying 2.8% deltamethrin WP insecticide (Scientific Fertiliser Co. PVT. Ltd, Mysuru, Puram Colinbatore, India) at a dose of 20mg a.i. /m$^2$ on the exterior walls of all huts ("rakobas"), and on both the interior and exterior surface of the household compound boundary fence. Restricted ODRS (RODRS), comprised of spraying only the boundary fence as described above. Control households and fences were left unsprayed.

Insecticide was applied using a 10 litre Hudson spray pump (Hudson Products, X-Pert 67322AD Professional Sprayer, Hudson Manufacturing Company, Hong Kong). The tank solution was prepared by one trained technician, and the insecticide applied by three trained Gedarif health authority personnel responsible for routine IRS under the regional IVM program. The interventions were supervised by the research team. All spraying personnel wore appropriate protective clothing, goggles and masks.

## Sand fly sampling

Sand fly traps were placed pre- and post-intervention in the four sentinel households per cluster. Sticky traps (ST) were made of standard A4 sheets of white paper coated with sesame oil. In a recent study, we showed that when placed horizontally on the ground, these traps had a significantly higher efficiency in sampling *P. orientalis* as compared to when placed vertically [45]. Therefore, sets of 10 sticky traps were laid horizontally (sticky-side up) on the ground in a line spaced 0.5 m apart; one set was positioned 30cm from the inside of the household compound boundary fence (labelled "outdoor" trap location). Another set of 10 STs were similarly positioned on the outside of the boundary fence (labelled "peridomestic" trap location) that faced the earthen street. STs were in position from 18:00 hrs and collected the following morning at 06:00 hrs. In addition, a single CDC light trap (LT) (Model 512, John Hock Company, USA) was set outdoors within the household compound, positioned approximately 1 meter from the compound boundary fence and suspended 0.5 meters from the ground from a tree branch or bamboo pole. LT placement and collection times were the same as for the STs.

A standard chemical knockdown (KD) technique [17] was used to determine the numbers of sand flies resting inside the household sleeping hut (Experiment 1 only). Between 06:00–07:00 hrs, the floor was covered with white cotton sheets, the room tightly sealed, and then spatially sprayed with 250 ml combination of 0.2% Tetramethrin, 0.025% Cyluthrin, 1.0% Piperonyl Butoxide, and 98.8% solvents and propellants (Flytex, Khartoum, Sudan). Fifteen minutes post spatial spraying; sand flies were carefully collected from the floor sheets for identification.

### Sand fly identification

Collected sand flies were washed in mild detergent solution, rinsed in distilled water and immediately stored in Eppendorf tubes containing 70% ethanol. Sand flies were initially sorted under a dissecting microscope to separate *Phlebotomus* from *Sergentomyia spp. Phlebotomus* specimens were then individually mounted in PVA medium (BioQuipp, USA) to inspect the spermatheca, pharyngeal armature and male genitalia under a binocular microscope to confirm species identification following relevant taxonomic keys [15, 46].

### Statistical analysis

The unit of study was the cluster-level *P. orientalis* trap count. Trap counts of male and female *P. orientalis* combined were standardised to represent numbers captured per trap per night ("trap-night"). The sum number caught by one set of 10 STs per night, or by one CDC light trap per night, was considered a trap-night, respectively.

The intervention effects in treated compared to in control clusters were estimated by fitting the standardised trap counts to negative binomial regression models, including variables describing the experimental design structure as appropriate including trap-type code, trap-site code, number of trap-nights, days from intervention, and a quadratic term (days$^2$) to account for non-linearity in outcome measures over follow-up time (Experiments 2 and 3). Study cluster ID was entered as a cluster term in the model, and pre-intervention (baseline) $\log_{10}$-transformed (+1) number of *P. orientalis* was used as the model offset parameter.

The potential variability in intervention effects between trap types and trap sites, and time (days and days$^2$) from intervention, were examined by model inclusion and testing of relevant interaction term(s). Effect estimates were expressed as incidence risk ratio (IRR) generated using post-estimation routines (LINCOM) in STATA.

Data from Experiments 2 and 3 were fitted to equivalent mixed effects (ME) models where cluster ID was treated as a random effect term, but this did not improve the model fits (Log-likelihood Ratio Test: $p > 0.05$), or the models failed to converge.

Data were analyzed using Stata v.15.1 software (StataCorp LP, College Station, TX).

### Insecticide delivery cost estimation

The costs of insecticide use for ODRS and RODRS were calculated based on the average inner and outer surface boundary fence area to be sprayed. This was estimated to be 175m$^2$ by measuring the convenience sample of 40 households treated in Experiment 1. For ODRS, this was added to the exterior wall surface area of 3 huts per household using an estimated average surface area of a typical single hut of 49.2 m$^2$. Insecticide usage for each of the 40 households was also recorded. The average costs of ODRS and RODRS per household were then calculated including the *pro rata* costs of skilled labour for three trained labourers (sprayers) and one technician (to prepare the Hudson tank solution) as practised in this study. The labour time to deliver RODRS was an average 30% less than that to deliver ODRS. For reporting, the total costs were converted using the 2016 official exchange rate of $1 USD = 6.6 Sudanese pounds (SDG).

## Results

### Trapping efficiency

Across the three experiments, a total of 10,813 *P. orientalis* sand flies were captured in 2,033 trap-nights, of which 33% were female flies (Table 1). Trap counts in clusters not treated with insecticide (pre- and post-intervention) indicated that households sampled in Experiment 2,

**Table 1. Summary of the total numbers of *P. orientalis* captured during the studies of effects of outdoor residual spraying of insecticide against the vector of visceral leishmaniasis in Gedarif state, Sudan.**

| Trap site | Trap type[1] | Total *P. orientalis* | Male *P. orientalis* | Female *P. orientalis* | n trap nights[1] |
|---|---|---|---|---|---|
| Pre-intervention Controls | | | | | |
| Inside sleeping huts | KD | 4 | 0 | 4 | 8 |
| Outdoor | LT | 480 | 346 | 134 | 80 |
| Outdoor | ST | 531 | 283 | 248 | 78 |
| Peridomestic | ST | 560 | 298 | 262 | 78 |
| Sum | | 1575 | 927 | 648 | 244 |
| Pre-intervention Treated ODRS/RODRS | | | | | |
| Inside sleeping huts | KD | 7 | 1 | 6 | 8 |
| Outdoor | LT | 697 | 455 | 242 | 112 |
| Outdoor | ST | 1022 | 563 | 459 | 110 |
| Peridomestic | ST | 907 | 486 | 421 | 110 |
| Sum | | 2633 | 1505 | 1128 | 340 |
| Post-intervention Controls | | | | | |
| Inside sleeping huts | KD | 13 | 3 | 10 | 8 |
| Outdoor | LT | 2056 | 1539 | 517 | 218 |
| Outdoor | ST | 1825 | 1364 | 461 | 188 |
| Peridomestic | ST | 1905 | 1428 | 477 | 188 |
| Sum | | 5799 | 4334 | 1465 | 602 |
| Post-intervention Treated ODRS/RODRS | | | | | |
| Inside sleeping huts | KD | 2 | 0 | 2 | 8 |
| Outdoor | LT | 162 | 111 | 51 | 305 |
| Outdoor | ST | 334 | 195 | 139 | 267 |
| Peridomestic | ST | 308 | 177 | 131 | 267 |
| Sum | | 806 | 483 | 323 | 847 |
| Totals | | 10813 | 7249 | 3564 | 2033 |

Pre- and post-intervention total numbers of *P. orientalis* and trap nights are aggregated across three experiments and presented according to treatment types and trapping methods.

[1] total numbers of trap nights represent the sum of chemical knockdown (KD) events inside sleeping huts; a single CDC light trap (LT) set per household at outdoor sites per night, and a set of 10 sticky traps (ST) set at each outdoor and peridomestic site per night. Each set of 10 STs is considered a single trap-night.

conducted in Umsalala village (in 2017), produced substantially higher *P. orientalis* trap counts than in Experiment 1 (in 2016) or in Experiment 3 (in 2017), both conducted in Jebel-Algana (Fig 3; S1 Table; S2 Table).

Differences in LT and ST counts ("trapping efficiency") within and between experiments were also variable (Table 1; S2 Table), though controlling for covariates, no consistent differences in the mean numbers captured by the two methods were detected pre-intervention cluster samples (Z = 0.19, P = 0.846, N = 568), or in post-intervention control cluster samples (Z = 0.58, P = 0.564, N = 594). However, due to inherent differences between ST and LT mechanisms, the intervention effects on sand fly numbers were calculated separately for each trap type (thus trap location).

## Experiment 1

The initial experiment was conducted to assess the feasibility of ODRS as a vector control option. *P. orientalis* were trapped 1–4 days pre-ODRS, and 1–4 days post-ODRS application. Accounting for study design characteristics, *P. orientalis* abundance was reduced by 83% (95%

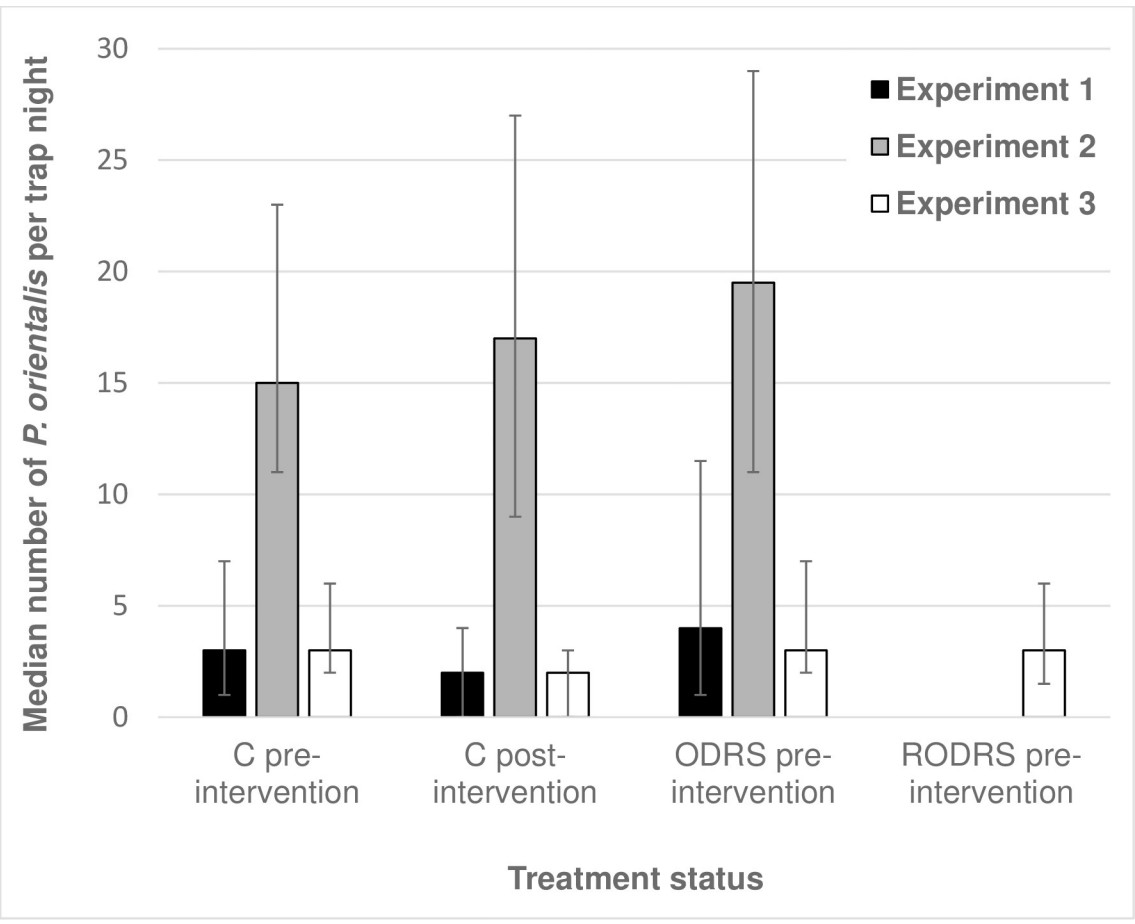

**Fig 3. Variations in the natural numbers of *P. orientalis* at households in Jebel-Algana and Umslala villages, Gedarif state, Sudan. Sand flies were captured by CDC light traps and sticky traps at untreated households before (pre-intervention) and after treatment (post-intervention).** Houses labelled by their post-intervention subsequent assignment to full (ODRS) or restricted (RODRS) outdoor residual insecticide spraying, or as untreated controls (C). Data are aggregated for the three independent experiments: Exp. 1 conducted in Jebel-Alagna, 9th-16th June 2016; Exp. 2 conducted in Umsalala village, 2nd May-16th June 2017; and Exp. 3 conducted in Jebel-Alagna, 31st March-20th June 2017. Error bars represent the interquartile range.

CL: 70.1%-90.9%) and 87% (95% CL: 76.3%, 93.0%) at outside and peridomestic ST locations, respectively, relative to in control clusters (Table 2, Fig 4).

   *P. orientalis* captures in sleeping huts collected by chemical knockdown (KD), and by LTs in outdoor locations, were few but showed significant reductions: inside huts, 4 and 7

**Table 2. Effects of outdoor residual spraying of insecticides (ODRS) on the numbers of *P. orientalis* in Jebel-Algna village, Gedarif state, Sudan. Sand fly trapping was conducted between 9th-16th June 2016.**

| Trap type[1] | Trap location[2] | IRR (95% C.L.) | P< | N Trap nights |
|---|---|---|---|---|
| ST | O | 0.17 (0.091, 0.299) | 0.0001 | 128 |
| ST | P | 0.13 (0.070, 0.237) | 0.0001 | 128 |

Effect estimates are presented as incidence risk ratios (IRR) according to trap type and trap location.

[1] Trap types for monitoring: ST sticky traps

[2] Trap site locations: O outdoors, P peridomestic.

ST captures at outdoor and peridomestic trap locations were not dissimilar (test of intervention × trap site interaction term: z = 1.53, p = 0.125, N = 256 trap nights) (Table 2).

(A)

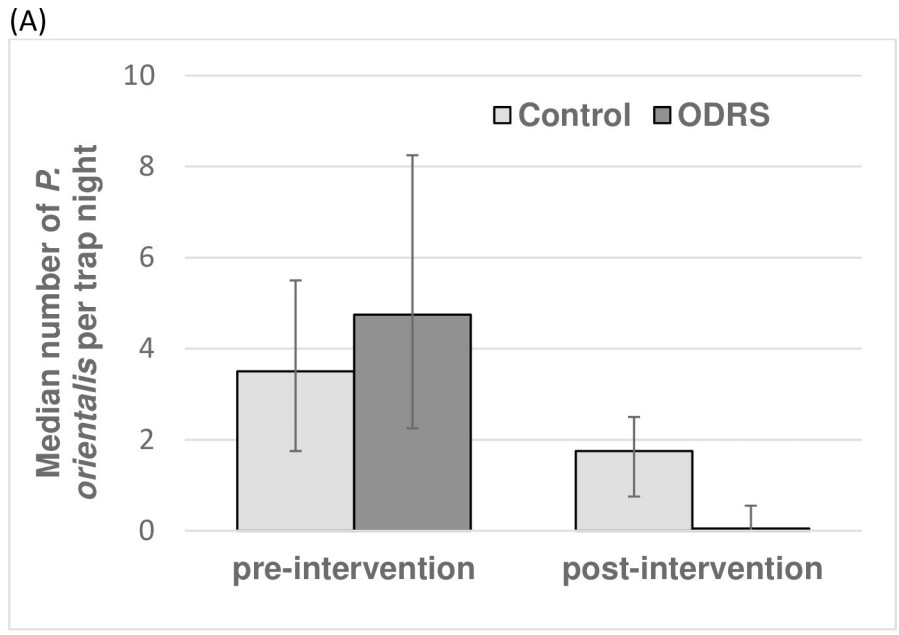

(B)

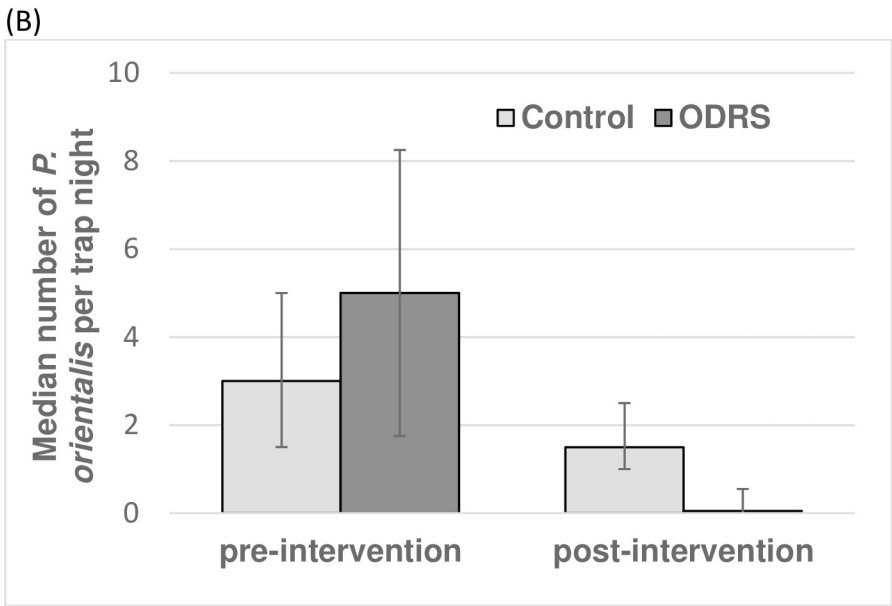

**Fig 4. Effects of outdoor residual spraying of insecticide (ODRS) on the numbers of *P. orientalis* sand flies in Jebel-Algana village, Gedarif state, Sudan.** Sand flies were captured by sticky traps (ST) set at outdoor sites (A), and at peridomestic sites (B) during 1–4 days pre-intervention and 1–4 days post-intervention. Each bar represents the median numbers of *P. orientalis* from 32 trap nights (i.e. 4 trap nights at 4 houses in two clusters each). Error bars represent the interquartile range. Sand fly trapping was conducted between 9th -16th June 2016.

specimens were collected in control and ODRS clusters respectively by pre-intervention sample, *versus* 13 and 2 specimens post intervention (Fisher's exact: p = 0.014; N = 16 trap nights). In outdoor LTs, 17 and 42 specimens were collected pre-intervention compared to 7 and 0 flies post-intervention (Fisher's Exact, P<0.001, N = 16 trap nights).

**Table 3. Effects of outdoor residual spraying of insecticide (ODRS) on the numbers of *P. orientalis* over 43 days follow-up between 2nd May-16th June 2017 in Umsalala village, Gedarif state, Sudan.**

| Trap type[1] | Trap local[2] | IRR (95%C.I.) | P< | N trap nights |
|---|---|---|---|---|
| ST | O | 0.04 (0.028, 0.080) | 0.001 | 180 |
| ST | P | 0.08 (0.051, 0.113) | 0.001 | 180 |
| LT | O | 0.01 (0.005, 0.008) | 0.001 | 238 |

Effect estimates are presented as incidence risk ratios (IRR) according to trap type and trap location

[1] Trap types for monitoring: ST sticky traps, LT light trap

[2] Trap site locations: O outdoors, P peridomestic.

## Experiment 2

The effects of a single ODRS application over 43 days follow-up was assessed. ODRS reduced ST numbers by an average 96% (95% CL: 92.0%, 97.2%) and 92% (95% CL: 88.7%, 94.9%) in outdoor and peridomestic locations, respectively, and by 99% (95% CL: 99.2%, 99.5%) in LT outdoor locations, relative to control clusters (Table 3; Fig 5). Few sand flies were captured in either intervention arm at 43 days post intervention (Fig 5).

Accounting for the trial design, the ODRS effect was significantly greater measured by the single LT, than by the set of 10 STs at outdoor trap locations (test of intervention × trap type interaction term: Z = 2.57, P = 0.010, N = 418). The effect estimate was also greater measured at outdoor ST compared to at peridomestic ST locations (test of intervention × ST site-code interaction term: z = 3.65, P = 0.001, N = 360) (Table 3).

## Experiment 3

The third experiment measured the insecticide effects over 76 days follow-up of a variant of ODRS, namely where insecticide application was restricted to only household boundary fences (RODRS). Since there was no difference in effect estimates between outdoors and peridomestic ST sites under either RODRS or ODRS interventions (test of intervention × trap location: Z<0.76, P>0.393), the data for these two trap locations were combined. Across ST sites, RODRS reduced vector numbers by a mean 60% (95% CL: 19.7, 80.5) compared to 85% (95% CL: 70.7%, 92.5%) by ODRS, both relative to in control clusters (Table 4; Fig 6). In outdoor LTs, the equivalent estimates were 88% (95% CL: 75.7%, 94.6%) and 93% (95% CL: 84.5%, 96.6%) respectively (Table 4; Fig 6).

Comparing RODRS to ODRS intervention outcomes directly (i.e. not compared to control clusters), the RODRS effect estimates on ST catches were on average 58% (95% CL: 43.6%, 74.1%, N = 446) lower than achieved by ODRS (Z = -4.09, P<0.001, N = 446). In contrast RODRS proved 100% as effective as ODRS when measured by outdoor LT captures (Z = 0.76, P = 0.449, N = 223).

## Economic costs of ODRS and RODRS delivery

The commercial cost of the insecticide was $15.15 USD per litre. Based on the total surface area to cover, the price per household was $3.49 USD for ODRS, and $1.89 USD for RODRS. The spray team delivered ODRS to 40 households in a single day. The joint labour costs for the three technicians was $90.91 USD per day, or $2.27 USD per household. To deliver RODRS, the labour costs fell to $1.59 USD per household. Excluding expenses of transport and administration, the relative sum costs of ODRS and RODRS delivery per household was 3.49+2.27 = $5.76 USD, and 1.89+1.59 = $3.48 USD, respectively. Thus, RODRS resulted in an average saving of 39.6% over ODRS.

(A)

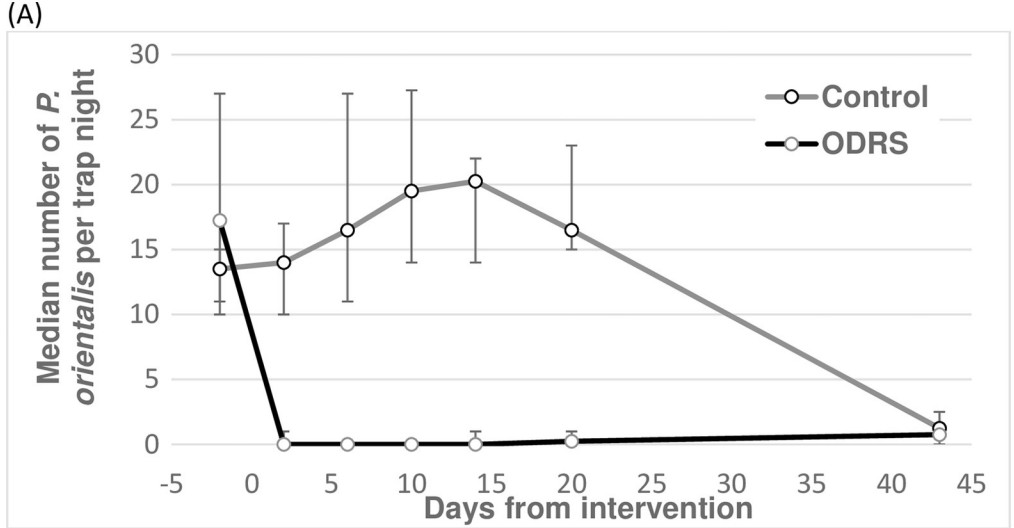

(B)

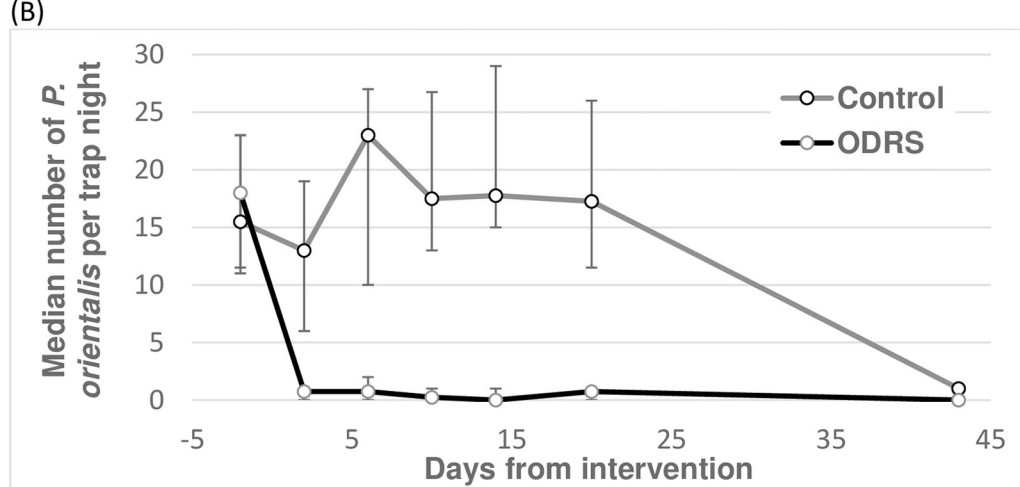

(C)

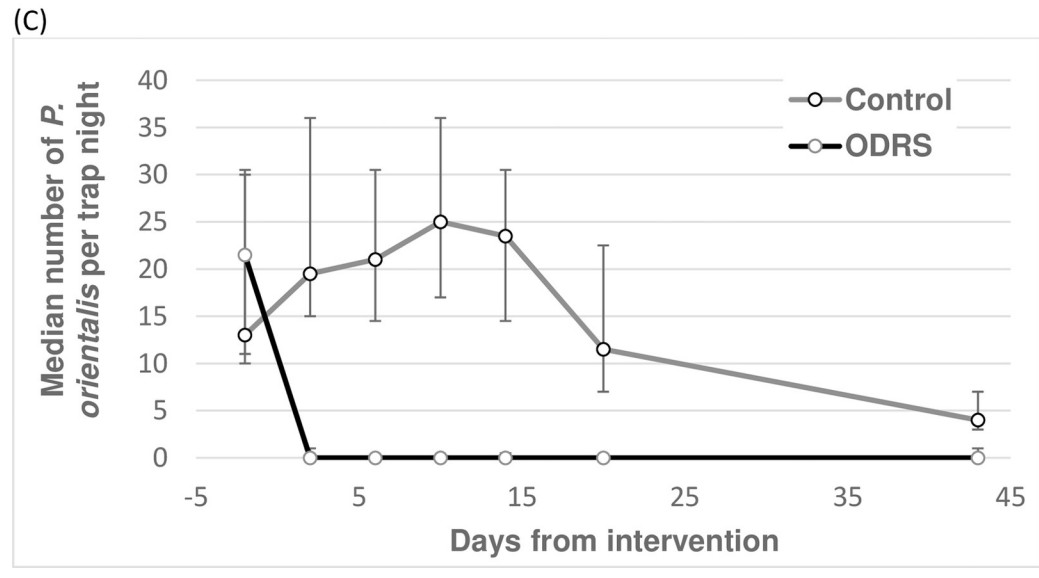

**Fig 5.** Effects of outdoor residual spraying of insecticides (ODRS) on the numbers of *P. orientalis* in Umsalala village, Gedarif state, Sudan. Sand flies were captured by sticky traps set at outdoor sites (A) and peridomestic sites (B), and by CDC light traps set at outdoor sites (C). Each point represents the median numbers of *P. orientalis* from 10–14 trap nights (A & B), or 15–16 trap nights with exception of 23–24 trap nights on day 43 (C). Error bars represent the interquartile range. Note differences in the Y-axis range. Sand fly trapping was conducted between 2nd May-16th June 2017.

## Discussion

The reported VL case incidence in East Africa (Sudan, South Sudan, Ethiopia, Somalia, Kenya, and Uganda) is now greater than that recorded in the Indian subcontinent, which was traditionally considered to suffer the heaviest VL burdens [5,33,47]. The lack of sustainable community-wide methods to reduce exophilic/exophagic disease vectors such as *P. orientalis* is a serious hindrance to efforts to reduce VL burdens in East Africa. This is despite recent international investment to scale-up early case detection and treatment as the only currently employed methods to combat the disease. At the time of study, the regional VL case incidence was 26–37/1000/year (2014–2017 MoH data), which is not dissimilar to published historical values of 38–39/1000/year (1991–1993) [35], 20–42/1000/year (1994–1996) [38], and 20–25/1000/year (2010–2011) [37] in the same region.

The results of the current study demonstrate that Outdoor Residual Spraying (ODRS) with pyrethroid insecticide can significantly reduce vector numbers in the outdoor compound, and in proximate peridomestic locations outside the boundary fence, where high vector numbers are known to occur [20,36]. Despite the variation in prevailing vector numbers between the two villages in this study, and the general seasonal decline in sand fly abundance, the intervention effects were consistent across experiments showing, importantly, that a single application of outdoor insecticide early in the transmission season could impact on outdoor and peridomestic sand fly vector numbers with lasting effects until the end of the transmission season, which starts in March and ends in June. Overall reductions of 83%-99% were achieved and remained more or less constant for the duration of the study period (maximum 76 days follow-up). The longitudinal experiments (Experiments 2 and 3) were conducted from April/May to June, representing a large fraction of the peak sand fly season (March to June) in east Sudan, NW Ethiopia and South Sudan. ODRS also reduced vector abundance in sleeping huts, though the numbers collected were few and insufficient to calculate robust IRR estimates.

In an attempt to reduce the quantity of insecticide and spraying effort required for ODRS, we investigated the entomological efficacy of RODRS by restricting spraying to the household boundary fence only (Experiment 3). The rationale for this alternative was based on prior observations of large numbers of vectors on the proximate exterior side of the fences. Relative to control clusters, RODRS caused an immediate suppression of outdoor and peridomestic *P. orientalis* abundance, with overall reductions during follow-up of 60% and 88% by ST and LT

**Table 4. Effects of full (ODRS) and restricted (RODRS) outdoor residual spraying of insecticide on the numbers of *P. orientalis* captured over 76 days follow-up between 1st April-17th June 2017 in Jebel-Algana village, Gedarif state, Sudan.**

| Treatment comparisons | IRR (95%C.I.) | P< | Trap type | Trap local | N trap nights |
|---|---|---|---|---|---|
| ODRS vs control | 0.15 (0.075, 0.293) | 0.001 | ST | O, P | 670 |
| RODRS vs control | 0.40 (0.195, 0.803) | 0.010 | ST | O, P | 670 |
| ODRS vs control | 0.07 (0.034, 0.155) | 0.001 | LT | O | 349 |
| RODRS vs control | 0.12 (0.054, 0.243) | 0.001 | LT | O | 349 |

Effect estimates are presented as incidence risk ratios (IRR) by trap type and trap location.

Trap types and trap site location abbreviations are as shown in Table 3.

(A)

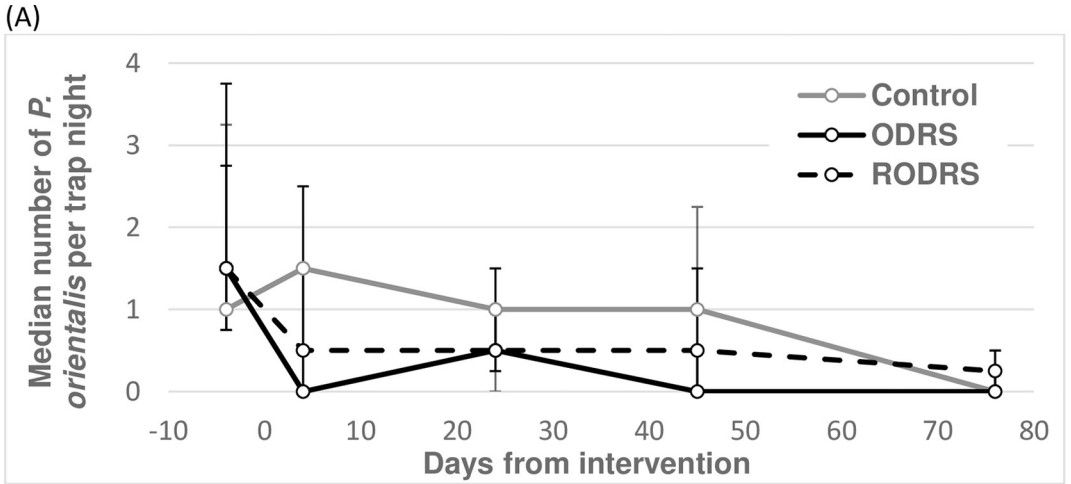

(B)

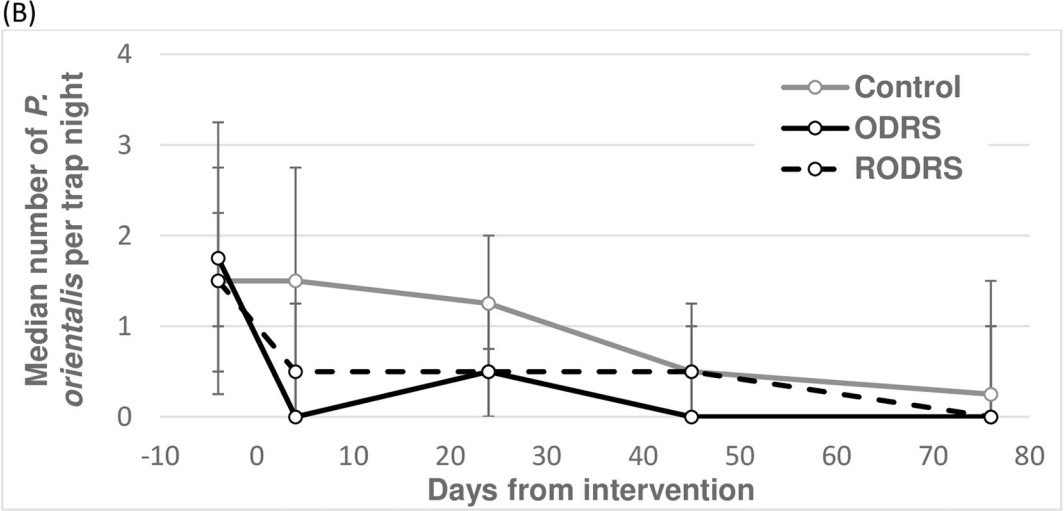

(C)

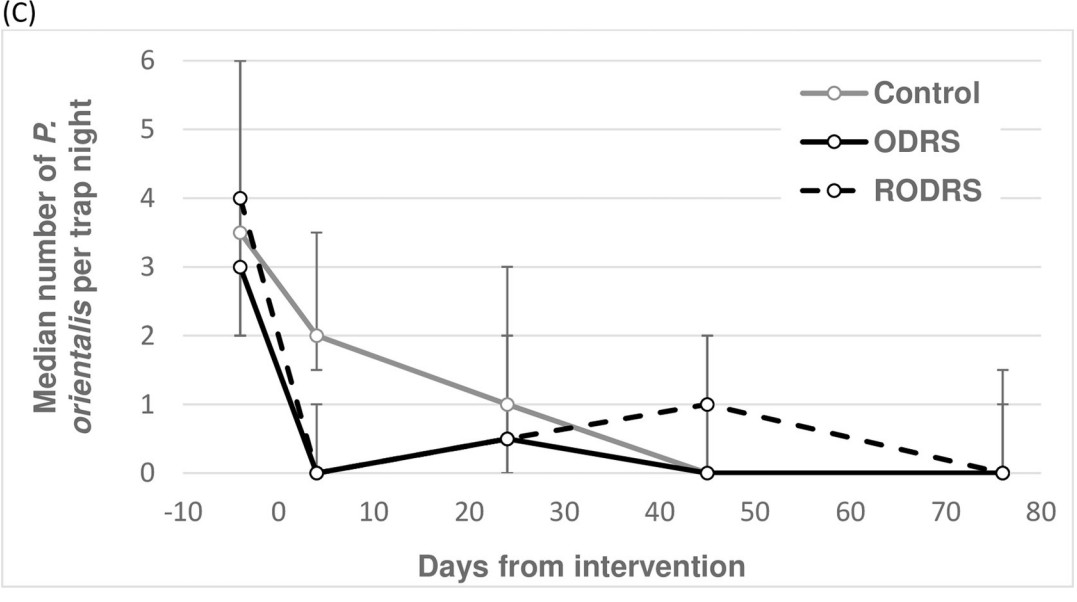

**Fig 6. Effects of full (ODRS) and restricted (RODRS) outdoor residual spraying of insecticide on the numbers of *P. orientalis* in Jebel-Algana village, Gedarif state, Sudan.** Sandflies were captured by sticky traps set at outdoor sites (A) and peridomestic sites (B) and by CDC light traps set at outdoor sites (C). Each point represents the median numbers of *P. orientalis* from 16 trap nights, except 32 trap nights on days -4 and +4, and 19–21 trap nights on day 76 in (C). Error bars represent the interquartile range. Note differences in the Y-axis range. Sand fly trapping was conducted between 31st March-20th June 2017.

estimation respectively. These values compared to 85% and by 93% for ODRS in the parallel treatment arm. Direct comparisons of RODRS and ODRS effect estimates (i.e. not compared to control clusters) gave mixed results: RODRS was an average 58% as effective as ODRS based on ST captures (P = 0.001), and 100% as effective based on LT captures (P>0.05). Here, *P. orientalis* numbers captured by the two trapping methods refer to a single LT located 0.5 m above ground, as compared to an individual ST comprising $10 \times$ A4 white paper sheets treated with sesame oil placed horizontal on the ground surface. We did not detect a consistent distinction in trap numbers (trapping efficiency) between LT and ST methods, or trapping success between outdoor *versus* peridomestic trap locations (S2 Table), by which to explain the inconsistency in relative effectiveness of RODRS. Additional experiments would be useful to increase the precision of the RODRS effect estimate.

The LT and ST methods used in the current study are standard tools for random sampling of sand flies [48]. To avoid the LT attraction of *P. orientalis* from outside the sentinel sites, we restricted the use of these traps to the courtyards of the houses, where presence of fences will prevent transmission of their light between clusters of houses. In contrast, we used the ST on both peridomestic and the inside of the houses. In the past, these traps were commonly placed vertically to intercept a presumed flight path of sand flies [48]. However, recently we demonstrated that for trapping *P. orientalis*, ST placed horizontally on the ground resulted in an 8-fold increase in sand fly numbers compared to located vertically [45]. Here we report the absolute numbers, rather than the density of, *P. orientalis* captured per trap night, which for ST consisted of a series of 10 ST in place from 18:00–06:00 h. If required, the numbers can be converted to a density per $m^2$ based on the surface area of the A4 paper (0.6237 $m^2$), thus requiring multiplication by 1.6 to represent the 10 ST.

The boundary fences sprayed in this study are 1.5–2 m high constructed of tightly woven reed, which surround individual household compounds to provide privacy. The fences are likely to act as a physical barrier to blood-seeking *P. orientalis* incoming from more distant peridomestic and sylvatic habitats. Their significance as a target for ODRS/RODRS can be explained by how sand fly locomotion behaviour facilitates contact with the non-repellent pyrethroid insecticide. Sand fly vector species tend to stay close to the ground at <1 m height, and traverse the ground, flat or sloping surfaces, by a series of "hop-like" motions [49]. Observations also show that when they encounter a vertical obstacle, they proceed upwards close to the obstacle, landing on the substrate in between short flight steps [49]. Our controlled experiments indicate that vectors indeed landed on the insecticide-treated fences resulting in a lethal insecticide dose causing significant reductions in sand fly abundance. Repellence (diversion) from treated surfaces was not observed.

Such boundary fences are common across the VL endemic villages in Gedarif, Sinnar and Blue Nile states, in addition to other endemic regions of Sudan, where ODRS / RODRS could be applied to control transmission. The findings may be relevant also to other high VL incidence regions of East Africa where *P. orientalis* is the vector, for example, amongst the vast populations of seasonal migrant agricultural workers e.g. on farms along the Sudan/Ethiopian border, where 60% of transmission of VL in Ethiopia occurs [21,50]. These notoriously mobile populations work in the agricultural farms during two periods; May-July to prepare land for planting crops, and September-November for harvesting [50]. During the VL transmission

season (May–July), the majority of workers sleep in small shelters made of grass [21]. We suggest that insecticide-impregnated fences erected around existing, or as newly built, night-time shelters could provide significant protection from the blood-seeking *P. orientalis*. In essence, insecticide sprayed fences represent sandfly barriers that can also be utilized to protect semi-nomadic populations such as cattle herders or agricultural farmers who tend to stay in temporary camping grounds.

One potential concern for the development of ODRS programmes is the expected short (e.g. 2–3 month) residuality of pyrethroids sprayed onto outdoor surfaces exposed to intense UV radiation [51]. Fortunately, the sand fly season in eastern Sudan is short (3 months) and abruptly ends with the heavy rains that occur at the end of June [17]. The heavy rains seals cracks in the soil where the vector is thought to reside [41]. The fall in the numbers of *P. orientalis* observed in unsprayed households, at the end of May 20–43 days in experiment 2 and 45–76 days in experiment 3, coincided with the beginning of the rainy season.

One shortcoming of the current study was that we did not test the residuality of the deltamethrin a.i. more formally by performing sand fly exposure bioassays and/ or quantitative chemical analyses (e.g. IQK [52] and/or HPLC). Such evaluations also would be informative.

## Sand fly control

At the time of writing, the IVM program in Sudan recommends IRS using pyrethroid or Bendiocarb insecticide applied twice per year (December and June), and distribution of long-lasting insecticide-impregnated bednets (LLINs, Vestergaard Frandsen, Lausanne, Switzerland) (Mr Anwar Osman Banaga Gedarif State Ministry of Health, pers. comm.). IRS and LLINs principally target endophilic vectors (modelled on endophilic malaria transmitting mosquitoes), thus precluding full effectiveness against this exophilic sand fly vector. Furthermore, IRS deployment does not coincide with the season of sand fly activity. The collective studies in Sudan and NW Ethiopia lead to the consensus that *P. orientalis* has a low propensity to enter household buildings [17,21,25,26,27,28,53,54], which is corroborated by the current data showing that comparatively few *P. orientalis* were captured inside sleeping huts as measured by chemical knockdown (KD) methods. To our knowledge, there are no peer-reviewed evaluations of IRS against *P. orientalis* in East Africa.

Previous attempts to reduce *P. orientalis* abundance and/or *L. donovani* transmission include ultra-low volume (ULV) fogging of *A. seyal* thickets [55], provision of insect repellents [20], and ITNs [9,56,57]. Whilst the community-wide distribution of ITNs was retrospectively associated with protection against VL under epidemic conditions [9], ITNs are not generally used in resident communities during the *P. orientalis* biting season which is characterised by particularly high temperatures. During this period, people habitually sleep outside in the compound without a bednet. ITNs tend to be used more inside huts during the rainy / cooler season (late June to October) when nuisance / malarial mosquito vectors are most abundant, but *P. orientalis* is no longer active. In a previous ITN study, which took place in Sudan, bednets were set up at 9-11pm, leaving children unprotected during a significant period of peak sand fly biting time at 6–9 pm [9]. Furthermore, the study reported bednet use by <10% of the sample population during the hot dry months (sand fly season), only rising to 55% during the beginning of the rainy season (end of sand fly season).

## Intervention costs

The relative costs of RODRS and ODRS were estimated to be USD $3.48 and USD $5.76 per household, respectively, with a saving of about 40% by RODRS *versus* ODRS. These represent simple calculations for labour and insecticide, assuming similar implementation costs

(transport, equipment and administration) for the two vector control strategies. A recent study in the region suggested that the cost of IRS using deltamethrin or Bendiocarb was about USD $2.2-$2.85 per person year, based on more sophisticated health economic models, concluding that IRS + LLINs were highly cost-effective for malaria control [34] as defined by WHO [58]. Direct cost comparisons with values calculated in the present study are not strictly valid, especially with variation in methods of insecticide procurement, and the rapidly declining exchange rate of the Sudanese pound. Nonetheless, considering 5 persons per endemic household in the region [9], ODRS and RODRS per person year would appear relatively low cost per household. Prevention is clearly less costly than VL management, with estimated direct and indirect costs of treatment for a single VL episode being $450 USD [59] The cost-benefit is likely to be even greater considering that a proportion of subsequent post-Kala azar dermal leishmaniasis (PKDL) cases would also be avoided. Governmental and non-governmental agencies provide treatment drugs free of charge, but households are reported to bear 53% of the total costs, accounting for 40% of the median annual household income in Gedarif state [59].

In conclusion, these pilot studies establish the feasibility and potential entomological effectiveness of ODRS and RODRS as novel vector control options in Gedarif state and other endemic regions of Sudan. A key finding was that the observed levels of impact were achieved by a single application of insecticide. These community-wide interventions are likely to have high levels of social acceptability as they address the compliance issues around bednet use, and the populations are already accustomed to insecticide spraying. Although the critical threshold in sand fly abundance to achieve protection against transmission and VL is not yet known, the collective results of these pilot studies suggest that ODRS/RODRS are strong candidates for a more sustainable community-wide vector control approach in endemic regions in Sudan, and possibly elsewhere where sand fly vectors are particularly exophilic and exophagic and where building such fences may be feasible. Intervention trials to assess the effectiveness and cost-effectiveness of this approach on *L. donovani* transmission and VL incidence are now warranted. In addition to monitoring disease incidence, *P. orientalis* abundance and man-biting rates, future trials should also assess the intervention effects on the infection rates of *L. donovani* in the vector.

## Supporting information

**S1 Table. Total numbers of *P. orientalis* captured in the three Experiments pre- and post-intervention according to trap type and trap location.**
(DOCX)

**S2 Table. Median (IQR) numbers of *Phlebotomus orientalis* sand flies captured per trap night before and after outdoor residual insecticide spraying of exterior walls of sleeping huts, and household boundary fences (ODRS), or boundary fences alone (RODRS), in Gedarif state, eastern Sudan.** (a) Experiment 1 conducted in June 2016 in Jebel-Algana village; (b) Experiment 2 conducted in May-June 2017 in Umsalala village; (c) Experiment-3 conducted in March-June 2017 in Jebel-Algana village. For Experiments 2 and 3, median values were calculated over 43- and 76-days follow-up respectively.
(DOCX)

## Acknowledgments

We would like to express our deepest gratitude to Dr. Naiema Algaseer, and other colleagues at the WHO office in Khartoum, the KalaCORE administration at Mott MacDonald (UK), the University of Khartoum, University of Gezeira, and University of Gedarif (Sudan) for

facilitating the logistics and the finance of the fieldwork. Thanks are also due to several field assistants who helped in the field study. Special thanks to Mr. Hag Omer and Mr. Ahmed Hussein and the population of Belo village, Gedarif state, for hosting us throughout the project. We are grateful to Dr M. Kairo (Dean of the School of Agricultural and Natural Sciences) and Dr J. Cumming (current chair of the Department of Natural Sciences, UMES), for their support during the study and the publication of the manuscript.

## Author Contributions

**Conceptualization:** Dia-Eldin A. Elnaiem, Orin Courtenay.

**Data curation:** Dia-Eldin A. Elnaiem, Osman Dakein.

**Formal analysis:** Dia-Eldin A. Elnaiem, Orin Courtenay.

**Funding acquisition:** Dia-Eldin A. Elnaiem, Margriet Den Boer, Koert Ritmeijer, Caryn Bern, Jorge Alvar.

**Investigation:** Dia-Eldin A. Elnaiem, Osman Dakein, Ahmed Mohammed-Ali Alawad, Bashir Alsharif, Altayeb Khogali, Tayseer Jibreel, Omran F. Osman, Hassan Has'san, Mousab Elhag, Noteila Khalid.

**Methodology:** Dia-Eldin A. Elnaiem, Orin Courtenay.

**Project administration:** Dia-Eldin A. Elnaiem, Omran F. Osman, Atia Mohamed Atia, Mousab Elhag.

**Resources:** Dia-Eldin A. Elnaiem.

**Supervision:** Dia-Eldin A. Elnaiem, Noteila Khalid.

**Validation:** Dia-Eldin A. Elnaiem.

**Writing – original draft:** Dia-Eldin A. Elnaiem, Orin Courtenay.

**Writing – review & editing:** Dia-Eldin A. Elnaiem, Margriet Den Boer, Koert Ritmeijer, Caryn Bern, Jorge Alvar, Noteila Khalid, Orin Courtenay.

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
