## [Decision Letter · Decision Letter 0]

29 Jul 2020

Dear Dr Elnaiem,

Thank you very much for submitting your manuscript "Outdoor Residual Insecticide Spraying, a New Approach for the Control of the Exophilic Vectors of Human Visceral Leishmaniasis: Phlebotomus orientalis in East Africa" for consideration at PLOS Neglected Tropical Diseases. As with all papers reviewed by the journal, your manuscript was reviewed by members of the editorial board and by several independent reviewers. In light of the reviews (below this email), we would like to invite the resubmission of a significantly-revised version that takes into account the reviewers' comments. 

Dear colleagues,

thank you for submitting your work for publication in PLoS NTD. As you will see, the reviews are generally favourable, but a revision of the manuscript will be needed.

We cannot make any decision about publication until we have seen the revised manuscript and your response to the reviewers' comments. Your revised manuscript is also likely to be sent to reviewers for further evaluation.

Sincerely,

Joachim Clos

Associate Editor

Shan Lv

Deputy Editor

Dear colleagues,

thank you for submitting your work for publication in PLoS NTD. As you will see, the reviews are generally favourable, but a revision of the manuscript will be needed.

Reviewer's Responses to Questions

**Key Review Criteria Required for Acceptance?**

**Methods**

-Are the objectives of the study clearly articulated with a clear testable hypothesis stated?

-Is the study design appropriate to address the stated objectives?

-Is the population clearly described and appropriate for the hypothesis being tested?

-Is the sample size sufficient to ensure adequate power to address the hypothesis being tested?

-Were correct statistical analysis used to support conclusions?

-Are there concerns about ethical or regulatory requirements being met?

Reviewer #1: Are the objectives of the study clearly articulated with a clear testable hypothesis stated? YES

-Is the study design appropriate to address the stated objectives? YES

-Is the population clearly described and appropriate for the hypothesis being tested? YES

-Is the sample size sufficient to ensure adequate power to address the hypothesis being tested? Limited but appropriate for preliminery studies

-Were correct statistical analysis used to support conclusions? I believe so but am no expert in statistical methods

-Are there concerns about ethical or regulatory requirements being met? NONE

Reviewer #2: The objectives of the study are clearly articulated and the study design is appropriate to address the stated objectives. The number and size of clusters in treated and control groups are sufficient to ensure adequate power to address the hypothesis being tested. A correct statistical analysis has been used to support conclusions. The type of study did not involve major ethical or regulatory requirements

Reviewer #3: (No Response)

**Results**

-Does the analysis presented match the analysis plan?

-Are the results clearly and completely presented?

-Are the figures (Tables, Images) of sufficient quality for clarity?

Reviewer #1: -Does the analysis presented match the analysis plan? Dont Know - seems sound methods were applied

-Are the results clearly and completely presented? YES BUT REQUIRE REVISION

-Are the figures (Tables, Images) of sufficient quality for clarity? REQUIRE WORK

Reviewer #2: Results are presented very clearly with the sufficient number of tables, images and graphs

Reviewer #3: (No Response)

**Conclusions**

-Are the conclusions supported by the data presented?

-Are the limitations of analysis clearly described?

-Do the authors discuss how these data can be helpful to advance our understanding of the topic under study?

-Is public health relevance addressed?

Reviewer #1: -Are the conclusions supported by the data presented? YES

-Are the limitations of analysis clearly described? REQUIRE CLARIFICATION

-Do the authors discuss how these data can be helpful to advance our understanding of the topic under study? YES

-Is public health relevance addressed? YES

Reviewer #2: The conclusions were supported by data and a few limitations associated with sand fly biology (Ph. orientalis) were pointed out. Treatment efficacy and cost analysis were made addressing the public health relevance of controlling visceral leishmaniasis in the Sudan, and in East Africa in general

Reviewer #3: (No Response)

**Editorial and Data Presentation Modifications?**

Reviewer #1: My comments inserted directly in the annotated PDF attached. Please share with Authors

Reviewer #2: The report is very well written and I have no minor modifications to suggest

Reviewer #3: (No Response)

**Summary and General Comments**

Reviewer #1: Sand fly control is a topic rarely addressed in the literature and even more rarely is it attempted in any meaningful and/or experimental approach. This group has performed the study under the most difficult conditions in the middle of nowhere in Sudan. They present clear results with applicable suggestions. While many questions remain, it is a simple and easily applicable inexpensive approach that utilized existing reed fences as natural obstacles. Sand flies tend to surmount such obstacles by hopping on them, thereby, exposing themselves to insecticide. Certainly worth publishing and expounding upon.

Reviewer #2: Despite the use of traditional methodology of sand fly control (type of insecticide, pre- and post-tretament evaluation based on sand fly counts) this report can be considered very innovative in the context of an exophilic behaviour of flying insects proven to transmit human pathogens.

Reviewer #3: (No Response)

PLOS authors have the option to publish the peer review history of their article (what does this mean?). If published, this will include your full peer review and any attached files.

Reviewer #1: No

Reviewer #2: No

Reviewer #3: No
---

## [Editor Report · Decision Letter 1]

4 Sep 2020

Dear Dr Elnaiem,

We are pleased to inform you that your manuscript 'Outdoor Residual Insecticide Spraying (ODRS), a New Approach for the Control of the Exophilic Vectors of Human Visceral Leishmaniasis: Phlebotomus orientalis in East Africa' has been provisionally accepted for publication in PLOS Neglected Tropical Diseases.

Best regards,

Joachim Clos

Associate Editor

Shan Lv

Deputy Editor

Thank you for submitting a revised version of your manuscript in which you addressed the points raised by the reviewers over the original submission.

---

## [Editor Report · Acceptance letter]

12 Oct 2020

Dear Dr. Elnaiem,

We are delighted to inform you that your manuscript, "Outdoor Residual Insecticide Spraying (ODRS), a New Approach for the Control of the Exophilic Vectors of Human Visceral Leishmaniasis: *Phlebotomus orientalis* in East Africa," has been formally accepted for publication in PLOS Neglected Tropical Diseases.

Best regards,

Shaden Kamhawi

co-Editor-in-Chief

Paul Brindley

co-Editor-in-Chief
